

# Fitness implications of sex-specific catch-up growth in *Nephila senegalensis*, a spider with extreme reversed SSD

Rainer Neumann, Nicole Ruppel[†] and Jutta M. Schneider

Zoologisches Institut, Biozentrum Grindel, Universität Hamburg, Hamburg, Germany
[†] Deceased.

## ABSTRACT

**Background**. Animal growth is often constrained by unfavourable conditions and divergences from optimal body size can be detrimental to an individual's fitness, particularly in species with determinate growth and a narrow time-frame for life-time reproduction. Growth restriction in early juvenile stages can later be compensated by means of plastic developmental responses, such as adaptive catch-up growth (the compensation of growth deficits through delayed development). Although sex differences regarding the mode and degree of growth compensation have been coherently predicted from sex-specific fitness payoffs, inconsistent results imply a need for further research. We used the African *Nephila senegalensis*, representing an extreme case of female-biased sexual size dimorphism (SSD), to study fitness implications of sex-specific growth compensation. We predicted effective catch-up growth in early food-restricted females to result in full compensation of growth deficits and a life-time fecundity (LTF) equivalent to unrestricted females. Based on a stronger trade-off between size-related benefits and costs of a delayed maturation, we expected less effective catch-up growth in males.

**Methods**. We tracked the development of over one thousand spiders in different feeding treatments, e.g., comprising a fixed period of early low feeding conditions followed by unrestricted feeding conditions, permanent unrestricted feeding conditions, or permanent low feeding conditions as a control. In a second experimental section, we assessed female fitness by measuring LTF in a subset of females. In addition, we tested whether compensatory development affected the reproductive lifespan in both sexes and analysed genotype-by-treatment interactions as a potential cause of variation in life-history traits.

**Results**. Both sexes delayed maturation to counteract early growth restriction, but only females achieved full compensation of adult body size. Female catch-up growth resulted in equivalent LTF compared to unrestricted females. We found significant interactions between experimental treatments and sex as well as between treatments and family lineage, suggesting that family-specific responses contribute to the unusually large variation of life-history traits in *Nephila* spiders. Our feeding treatments had no effect on the reproductive lifespan in either sex.

**Discussion**. Our findings are in line with predictions of life-history theory and corroborate strong fecundity selection to result in full female growth compensation. Males showed incomplete growth compensation despite a delayed development, indicating relaxed selection on large size and a stronger trade-off between late maturation and size-related benefits. We suggest that moderate catch-up growth in males is still adaptive as

Corresponding author
Rainer Neumann, epeira@web.de

a 'bet-hedging' strategy to disperse unavoidable costs between life-history traits affected by early growth restriction (the duration of development and adult size).

## INTRODUCTION

Body size and the duration of development are among the most fitness-relevant life-history traits and have been extensively studied in various animal taxa (reviewed in *Blanckenhorn, 2005*; *Nylin & Gotthard, 1998*; *Roff, 2002*). Generally constrained by a trade-off between a favourable size at the onset of reproduction and the time necessary to reach it (*Blanckenhorn, 2000*; *Roff, 1992*), animal growth depends on both inherited growth trajectories and plastic modifications of them (*Chase, 1999*; *Dmitriew, 2011*). Extrinsic factors that may influence the mode of development include parasite infestations and other pathogens (*Paez, Fleming-Davies & Dwyer, 2015*; *Vergauwen et al., 2011*), cues of present or future environmental conditions (*Kasumovic & Brooks, 2011*), ambient temperature (*Kingsolver, Izem & Ragland, 2004*), and food supply (*Stearns, 1992*; *Wilson & Osbourn, 1960*).

Growth strategies and optimal body size may differ markedly between the sexes, which is particularly evident in sexually size-dimorphic species (*Blanckenhorn, 2005*). As large males tend to succeed in male-male competition across animal taxa, body size in males is often subject to sexual selection, resulting in male-biased sexual size dimorphism (SSD) (*Kingsolver & Pfennig, 2004*). Still enigmatic conditions are found in species with female-biased SSD, which mainly occurs in oviparous animals, like fishes (*Barreto, Moreira & Carvalho, 2003*), amphibians (*Hector, Bishop & Nakagawa, 2012*; *Nali et al., 2014*), and many invertebrates (*Honek, 1993*; *Smith & Brockmann, 2014*) including spiders (*Cheng & Kuntner, 2015*; *Foellmer & Moya-Larano, 2007*; *Higgins et al., 2011*; *Schneider & Andrade, 2011*). As females are strongly selected to produce large numbers of eggs, increased female size through fecundity selection is generally well supported in these species (*Blanckenhorn, 2005*; *Nylin & Gotthard, 1998*). Males, however, apparently are selected to stay small, which has been related to improved mobility and agility during mate search (*Moya-Laraño, Halaj & Wise, 2002*; *Moya-Laraño et al., 2009*), a decreased risk of predation and female sexual cannibalism (*Foellmer & Fairbairn, 2004*), benefits of protandry through rapid maturation (*Blanckenhorn et al., 2007*), and reduced energy expenditures (*Blanckenhorn, 2000*; *Blanckenhorn, 2005*).

Optimization of development and growth may be difficult in non-constant environments. Basic strategies that permit fitness maximization under invariant conditions may need to be refined in response to environmental changes (*Foster & Kreitzman, 2009*). While alterations of environmental conditions often appear as recurring sequences, irregular fluctuations of environmental parameters are also common in a range of habitats. Such unpredictable conditions pose a threat to an individual's fitness, particularly in species in which reproductive success depends on a single reproductive period (*Abrams et al., 1996*).

For example, unfavourable temperatures or food restriction may result in a delay of development, which can increase juvenile predation risk, but also lower reproductive prospects in individuals that reach maturity too late (end-of-season penalty; e.g., *De Block, McPeek & Stoks, 2008*; *Higgins, 2000*).

Phenotypic plasticity (the capacity of a genotype to express different phenotypes in different environments; *West-Eberhard, 2003*) provides the potential to counteract a period of unfavourable growth conditions in juvenile stages by means of adaptive developmental responses, e.g., compensatory growth and catch-up growth (*Dmitriew, 2011*; *Krause & Caspers, 2016*; *Metcalfe & Monaghan, 2001*; *Walzer, Lepp & Schausberger, 2015*). Compensatory growth refers to elevated growth under improved conditions, whereby a delay of sexual maturation is minimized. In contrast, adaptive catch-up growth is defined as a strategy to reach a favourable adult size at the expense of delaying maturation (*Hector & Nakagawa, 2012*; *Livingston, Kahn & Jennions, 2014*). Compensatory developmental mechanisms enable animals to either limit or entirely prevent fitness costs that would follow from a period of adverse growth conditions without the capacity for such flexible responses. However, early food restriction and subsequent growth compensation may entail intrinsic costs that can even lead to a reduction in lifespan (*English & Uller, 2016*; *Hornick et al., 2000*; *Reichert et al., 2015*).

As benefits and costs of developmental compensation may differ considerably between males and females, integrating pre-estimated divergent selection on body size may substantiate experimental work on such strategies. Plastic modifications of life-history traits have been related to experimental feeding regimes in a range of studies (*Bauerfeind & Fischer, 2005*; *Bonneaud et al., 2016*; *Dahl et al., 2012*; *Davidowitz, D'Amico & Nijhout, 2003*; *Fernandez-Montraveta & Moya-Larano, 2007*; *Kleinteich & Schneider, 2011*; *Krause & Caspers, 2016*), but relatively few of these have addressed sex-specific differences regarding compensatory development (*Arnold et al., 2007*; *Chin et al., 2013*; *Tawes & Kelly, 2016*).

In general, predictions concerning such differences are based on (1) proposed sex-specific net benefits of growth compensation (i.e., the sex whose fitness depends stronger on large body size is expected to show a higher degree of growth compensation), (2) on the possibility to increase size after sexual maturation (i.e., determinate versus indeterminate growth; with determinate growth generating stronger selection pressure to compensate growth deficits), and (3) on potential long-term costs of compensatory development (*Livingston, Kahn & Jennions, 2014*). Previous studies, however, are inconsistent as to whether predictions were met or not (*Barreto, Moreira & Carvalho, 2003*; *Livingston, Kahn & Jennions, 2014*; *Stillwell & Davidowitz, 2010*; *Tawes & Kelly, 2016*), thus indicating that possible trade-offs between growth compensation, taxon-dependent life-history, and environmental conditions that determine the adaptive value of compensation require further research.

Species showing strong SSD are particularly suitable model systems to investigate sex-specific compensatory mechanisms, because especially pronounced sex-differences concerning size selection can be comparatively studied in a single species. Golden-silk spiders (genus *Nephila*, family Araneidae) show some of the most extreme cases of female-biased SSD (*Kuntner et al., 2013*). Male and female size in these spiders has been suggested to have evolved independently, with steady fecundity-driven selection on increased female

size, whereas phylogenetic analyses did not reveal a consistent evolutionary trend towards male size-reduction (*Higgins et al., 2011*; *Kuntner & Elgar, 2014*). In addition, several studies have reported large male advantages in the context of mating (*Christenson & Goist, 1979*; *Elgar, Bruce & De Crespigny, 2003*; *Elgar & Fahey, 1996*; *Rittschof, 2010*). Causes of small male size remain thus ambiguous, which also applies to the remarkable within-sex size variation in many species (*Elgar & Fahey, 1996*; *Higgins et al., 2011*; *Schneider & Elgar, 2005*). *Nephila* spiders are short-lived animals with determinate growth (*Fromhage, Jacobs & Schneider, 2007*; *Miyashita, 2005*; *Rittschof, 2011*) and individuals of both sexes mature and reproduce within a limited time frame and within their own cohort (*Higgins, 2000*; *Higgins et al., 2011*).

Natural populations exposed to different environmental conditions have been studied in the American *N. clavipes* (*Higgins, 1993*; *Higgins, 1992*). This species is bivoltine in some populations, where first-generation females mature at larger average size than second-generation females. However, first-generation females pass through early development in the dry season, experiencing low feeding success and hence unfavourable juvenile growth conditions. These females may have adaptively delayed maturation, later taking advantage of improving feeding conditions to eventually mature at large size (*Higgins, 1992*). While these observations hint towards catch-up growth in females, experimental work is needed to test predictions following from this mechanism; specifically in comparison with permanently food-restricted and permanently well-fed individuals. Furthermore, the inclusion of males is essential to access sex-specific differences in an integral procedure to understand selection in this system.

Here, we consider the above-mentioned observations in the light of current research focussing on plastic compensatory mechanisms in sexually size-dimorphic species (*Chin et al., 2013*; *Kahn, Livingston & Jennions, 2012*; *Livingston, Kahn & Jennions, 2014*). Using the African *N. senegalensis*, we implemented a comprehensive approach consisting of two successive experimental sections. In the first section, we manipulated the study animals' feeding conditions, rearing split broods under constant low or high food supply, or in treatments in which the food supply was reciprocally reversed at a fixed point in time. Based on strong fecundity-selection for large female size (*Higgins & Goodnight, 2011*; *Kuntner et al., 2012*), we predicted effective catch-up growth after treatment reversal in initially food-restricted females to compensate the preceding deficits. As *Nephila* males generally benefit from protandry (*Danielson-Francois et al., 2012*; *Kasumovic et al., 2009*), selection should act against an exceedingly delayed development in males. Furthermore, flexible mating strategies have been found to balance reproductive success between differently-sized competitors (*Neumann & Schneider, 2015*). Therefore, we assumed weaker selection on large male size and predicted less effective catch-up growth in males.

Following the rearing treatments, we used a subset of adult females to measure life-time fecundity (LTF), thereby providing a direct test concerning the adaptive significance of growth compensation, which is often omitted in empirical studies (*Dmitriew, 2011*; *Hector & Nakagawa, 2012*). We predicted growth compensation to result in equivalent numbers of offspring in initially food-restricted females compared to constantly well-fed females.

As another measure of fitness, we tested whether growth compensation affected the post-maturation lifespan and hence the potential time-frame of reproduction in both sexes. Finally, we report treatment-related mortality and analyse genotype-by-treatment interactions as a potential cause of the remarkable variation of life-history traits in our model system.

## MATERIAL & METHODS

### Developmental duration, body size and weight, pre-maturation mortality

We collected eight gravid females near Cradock, Eastern Cape, South Africa, in March, 2008. Field-collected females were transferred to the laboratory and housed individually in 60 × 60 cm-sized Perspex frames. We maintained females under standardized conditions and all of them built viable egg sacs that were incubated in air-vented plastic containers until the offspring hatched (see *Schneider et al., 2011* for our standard methods concerning housing, feeding, and watering of spiders as well as temperature conditions). The hatchlings were separated at very small body size (2–4 mm) before they had reached the third instar (the first two moults in *Nephila* spiders occur inside the egg sac). Separated spiders were housed in small plastic cups but were transferred to larger cups as they increased in body size. We haphazardly allocated the study animals to the following feeding treatments: (1) High-High, (2) High-Low, (3) Low-High, and (4), Low-Low. Equal numbers of spiders from individual maternal lineages were used in each treatment. Spiders in the High-High treatment were provided with *ad libitum* food over the entire duration of development to maturity, whereas spiders in treatment Low-Low were kept at low-food conditions throughout the experiment. Study animals in treatment High-Low received *ad libitum* food during a fixed period of four weeks (defined as early experimental conditions) but were kept under low-food conditions in the period following the first four weeks (defined as late experimental conditions). The inversed pattern was adopted in treatment Low-High. The spiders were fed *Drosophila* and *Calliphora* flies. Low-food conditions conform to four *Drosophila* flies per week during early experimental conditions and 6–10 *Drosophila* flies (depending on the spider's size) per week during late experimental conditions, respectively. Low-food spiders large enough to eat *Calliphora* received two flies per week. *Drosophila* flies were raised on Carolina Biological Supply instant *Drosophila* medium Formula 4–24, which was enriched with additional nutrients, especially protein and vitamins. For this purpose, we mixed the medium with commercial high quality dog food according to a study by *Mayntz & Toft (2001)*, which demonstrated positive effects on growth and survival in a wolf spider fed with flies cultivated on this specific mixture. *Calliphora* flies were obtained by incubating fully grown larvae purchased from a commercial supplier. All study animals were reared in a daylight lab and hence were exposed to slight photoperiod changes. We checked the spiders on five days per week and tracked the development of each individual by recording the following data: sex, duration of development from the start of the experiment to maturation, weight at completion of the early experimental period, adult weight, and adult body size (given as patella-tibia length).

As morphology-based sex-determination is impossible in small juvenile spiders, individuals could be sexed only at larger developmental stages in the late experimental period, but 85 spiders died unsexed. Immature males were identified by their swollen pedipalps indicating the ongoing transformation into copulatory organs; the lack of this trait in juveniles with a body length ≥ approximately 12 mm indicated female sex.

## Post-maturation longevity

In addition to developmental modifications, we analysed treatment-related effects on adult longevity. For this purpose, we chose 137 males and 251 females across treatments upon reaching maturity. The spiders were maintained on our regular laboratory feeding schedule irrespective of the developmental feeding treatment experienced before. Spiders were chosen randomly; paying attention, however, to exclude animals to be used in the mating experiments (see 'Life-time fecundity and hatching success') or in our general breeding schedule. The remaining study animals were killed by hypothermia after reaching maturity and preserved at −80 °C. In total, 1,280 spiders were used in this study, of which 30 disappeared and another three were accidently killed during daily routine at early juvenile stages.

## Life-time fecundity and hatching success

We randomly chose 38 adult females originating from the treatments High-High ($N = 14$), Low-High ($N = 11$), and Low-Low ($N = 13$) to investigate whether compensatory growth enables females to overcome a period of poor feeding conditions during juvenile development and achieve a reproductive outcome equivalent to constantly well-fed females. We did not include High-Low females in this experiment (females in treatments High-Low and Low-Low did not differ significantly in developmental duration and size; see results). Females were maintained on our regular feeding schedule. We randomly chose adult males from the High-High feeding treatment to arrange mating trials. Prior to mating, each female was transferred to a Perspex frame (measuring $60 \times 60 \times 12$ cm) and given at least one day to build an orb-web, which is necessary for courtship and mating to take place. We positioned an unrelated male on the upper frame threads of the web. Each virgin couple was allowed to copulate once in a predefined period of 3 h. If copulation did not occur within the given time, we excluded the male from the study and arranged a second mating trial with a different male at a later date (each male was used only once). Six females remained unmated after the second trial and were excluded from the study.

At the beginning of each trial, the female received one *Calliphora* fly (males prefer mating with feeding females; *Schneider et al., 2011*). We measured the duration of copulation and removed the spiders from the web afterwards. Females were maintained in Perspex frames to build egg sacs until they died of age. Four females did not build egg sacs at all, despite apparently normal copulations. The egg sacs produced were incubated in air-vented plastic containers and preserved in alcohol after approximately five weeks. We carefully opened each egg sac and assessed the number of normally developed spiderlings, undeveloped eggs, and total clutch size. All experiments were carried out at the Zoological Institute, University of Hamburg.

## Statistical analyses

The study animals originated from eight maternal lineages from which we allocated equal numbers of individuals to each rearing-treatment. As a premise for further analysis, we tested if family lineages were evenly distributed among individuals that had passed through the feeding treatments and finally matured. The test confirmed no significant differences in the numbers of individuals originating from different family lineages between our feeding treatments ($G$-tests: males: $\chi^2 = 15.16$, $P = 0.82$; $N = 362$; females: $\chi^2 = 19.24$, $P = 0.57$; $N = 559$). We tested predictions with respect to adult body size, body mass, developmental durations, life-time fecundity (LTF), and post-maturation longevity using $t$-tests for normal data with equal variances and non-parametric Wilcoxon or Kruskal-Wallis tests if data diverged from these assumptions (indicated by Shapiro- and Bartlett's tests). Results are given as means $\pm$ SE, providing medians and interquartile ranges (IQRs) for non-parametric tests. Complementary post-hoc analyses were performed using pairwise Steel-Dwass tests that correct for Type I error inflation in multiple comparisons. We performed three linear mixed models to analyse sex-specific and family-related effects of our feeding treatments on development and growth. We tested effects of the variables Early treatment (High or Low), Family lineage, and Sex on body mass after completion of the early experimental period with a model containing all three variables and the interactions between Early Treatment and Family linage as well as between Early treatment and Sex. Models on total developmental duration (beginning with the day on which hatchlings were separated and allocated to individual treatments) and adult body size were equally specified with Treatment (early and late conditions), Family, and Sex, as well as the respective interaction terms. The variable Start date (the day of allocation to experimental treatments) was entered into the models as a random effect to account for a potential influence of photoperiod on development (Start date had nine levels ranging from June 11 to July 25, 2008). Body mass and developmental duration were log-transformed to improve the fit of the models. We analysed variation in LTF with a standard least-square model containing the interaction between Treatment and the number of clutches produced. All analyses in this study were conducted in JMP Pro 13 (SAS Institute Inc., Carey, NC, USA). Effect tests for individual variables in JMP are based on ANOVA-model comparisons between the full model and a reduced model lacking the respective variable. Additional statistical tests are denoted in the results section. Sample sizes within experiments may differ due to missing data.

## RESULTS

### Implications of early experimental conditions

Mortality rates during the first four weeks of the experiment did not differ significantly between early treatments; 59 of 626 spiders died under high feeding conditions (9.4%) and 74 of 630 spiders died under low feeding conditions (11.8%) ($G$-test: $\chi^2 = 1.79$, $P = 0.18$; $N = 1256$). (Sex-determination is impossible in small juvenile spiders; hence mortality rates were analysed independent of sex.)

In both sexes, body mass after the first four weeks differed considerably between early low and high feeding conditions. Males weighed $2.6 \pm 0.11$ mg at the end of the early low

**Table 1** **Effects of interactions between the variables Early treatment, Family lineage, and Sex on body mass at completion of the early experiment.** Results derive from a linear mixed model including the variable Start date (the time of allocating hatchlings to individual treatments) as a random effect to account for a potential influence of photoperiod on development. Body mass was log-transformed. Significant $P$ values are shown in bold.

| Explanatory variable | Body mass at completion of the early experiment | |
|---|---|---|
| | $F$ | $P$ |
| Early treatment | 1822.1 | **<0.0001** |
| Family lineage | 6.5 | **<0.0001** |
| Sex | 427.6 | **<0.0001** |
| Early treatment * Family lineage | 8.6 | **<0.0001** |
| Early treatment * Sex | 45.5 | **<0.0001** |

**Table 2** **Pre-maturation mortality compared between feeding treatments.** Treatment pairs were compared using $G$-tests. Significant $P$-values are shown in bold.

| Treatment pair (% mortality) | $\chi^2$ | $P$ | $N$ |
|---|---|---|---|
| High-Low (21%)–Low-Low (24.7%) | 1.044 | 0.307 | 569 |
| High-Low (21%)–High-High (9.3%) | 15.393 | **<0.0001** | 564 |
| High-High (9.3%)–Low-High (13.9%) | 2.816 | 0.093 | 545 |
| High-High (9.3%)–Low-Low (24.7%) | 24.156 | **<0.0001** | 563 |
| Low-High (13.9%)–Low-Low (24.7%) | 10.264 | **0.001** | 550 |
| High-Low (21%)–Low-High (13.9%) | 4.886 | **0.027** | 551 |

food period (median = 2.35 mg, IQR = 1.65); significantly less than those reared under early high feeding conditions, weighing $9.56 \pm 0.43$ mg (median = 8.26 mg, IQR = 6.46) (Wilcoxon test: $Z = 13.96$, $P < 0.0001$; $N = 363$). In females, early low feeding conditions resulted in a body mass of $4.08 \pm 0.12$ mg (median = 3.84 mg, IQR = 2.15), whereas early high-feeding females weighed $23.95 \pm 0.68$ mg (median = 22.25 mg, IQR = 17) (Wilcoxon test: $Z = 21.26$, $P < 0.0001$; $N = 665$). We conducted a linear mixed model to test for sex-specificity of juvenile growth as well as potential family-relatedness of growth responses. The model revealed highly significant effects of the interactions between Early feeding treatment and Family lineage as well as between Early treatment and Sex on the spiders' body mass at completion of the early experiment (Table 1).

## Implications of full experimental conditions
### Pre-maturation mortality

We analysed pre-maturation mortality in spiders that survived to experience both early and late experimental conditions and found distinctly different mortality rates ($G$-test: $\chi^2 = 29.1$, $P < 0.0001$; $N = 1114$; Table 2).

As mortality rates during the first four weeks did not differ between early high and low feeding conditions (see above), mortality differences concerning full experimental conditions may have solely been caused by late conditions. Pairwise between-treatment comparisons corroborated this assumption; irrespective of early feeding conditions, pre-maturation mortality rates differed significantly in treatment pairs exhibiting different

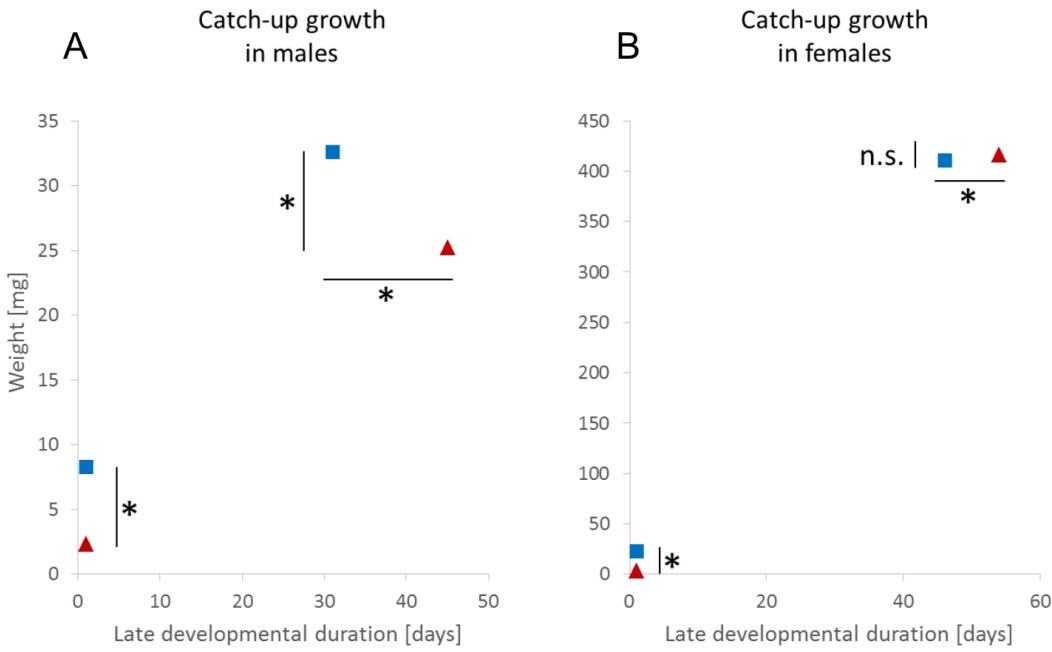

**Figure 1** **Adaptive catch-up growth in (A) male and (B) female *Nephila senegalensis*.** Symbols indicate median values for body mass at the beginning of the late-experiment development and body mass at sexual maturation in treatments High-High (blue squares) and Low-High (red triangles). Late-experiment development in the High-High treatment followed four weeks of early high feeding conditions. Late-experiment development in the Low-High treatment followed four weeks of early low feeding conditions. Body mass is given as a proxy of body size (*, indicates significant differences).

late feeding conditions, but not in treatment pairs in which late feeding conditions were identical (Table 2).

### Adaptive catch-up growth

In accordance with our predictions, late developmental durations in Low-High females significantly exceeded those of females in the High-High treatment (Late female developmental duration L-H: mean = 57.39 ± 1.04 days, median = 54 days, IQR = 8; H-H: mean = 46.72 ± 0.59 days, median = 46 days, IQR = 8.75; Wilcoxon test: $Z = 9.81$, $P < 0.0001$; $N = 292$). This delay of development resulted in full compensation of adverse early feeding conditions (Fig. 1). Females matured at similar size and weight in both treatments (Female adult size L-H: mean = 12.24 ± 0.13 mm, median = 12.22 mm, IQR = 1.9; H-H: mean = 12.45 ± 0.59 mm, median = 12.32 mm, IQR = 1.39; Wilcoxon test: $Z = 0.81$, $P = 0.416$; $N = 283$; Female adult weight L-H: mean = 433.84 ± 14.07 mg, median = 416.7 mg, IQR = 174.73; H-H: mean = 441.7 ± 12.6 mg, median = 410.83 mg, IQR = 159.67; Wilcoxon test: $Z = 0.22$, $P = 0.826$; $N = 282$).

Males in the Low-High treatment also delayed development compared to High-High males (Late male developmental duration L-H: mean = 45.46 ± 1.16 days, median = 45 days, IQR = 15; H-H: mean = 33.11 ± 1.05 days, median = 31 days, IQR = 11; Wilcoxon test: $Z = 7.56$, $P < 0.0001$; $N = 190$). In contrast to females, however, prolonged

**Table 3 Total developmental durations and adult size compared between feeding treatments.** Treatments were compared with Steel-Dwass pairwise tests (excluding treatment pair High-High–Low-High subject to predefined analysis of catch-up growth; see results). Significant $P$-values are shown in bold.

| Treatment pair | Males | | | | Females | | | |
|---|---|---|---|---|---|---|---|---|
| | Total duration of development | | Adult body size | | Total duration of development | | Adult body size | |
| | $Z$ | $P$ (N) | $Z$ | $P$ (N) | $\chi^2$ | $P$ (N) | $Z$ | $P$ (N) |
| High-High–Low-Low | 10.42 | <**0.0001** (184) | 9.06 | <**0.0001** (181) | 284.84 | <**0.0001** (283) | 13.55 | <**0.0001** (275) |
| High-High–High-Low | 7.17 | <**0.0001** (182) | 6.86 | <**0.0001** (179) | 226.58 | <**0.0001** (296) | 13.74 | <**0.0001** (285) |
| Low-Low–High-Low | 7.24 | <**0.0001** (172) | 5.32 | **0.0008** (168) | 4.67 | 0.2 (267) | 2.11 | 0.151 (262) |
| Low-Low–Low-High | 6.47 | <**0.0001** (180) | 4.92 | <**0.0001** (173) | 108.55 | <**0.0001** (263) | 12.26 | <**0.0001** (260) |
| High-Low–Low-High | 0.15 | 0.999 (178) | 0.95 | 0.776 (171) | 71.9 | <**0.0001** (276) | 12.14 | <**0.0001** (270) |

development did not fully compensate differences in male adult size and body mass (Male adult size L-H: mean = 4.95 ± 0.11 mm; H-H: mean = 5.62 ± 0.09 mm; $t$-test: $t = 4.86$, $P < 0.0001$; $N = 184$; Male adult weight L-H: mean = 26.59 ±1.09 mg, median = 25.25 mg, IQR = 13.81; H-H: mean = 34.23 ± 1.03 mg, median = 32.57 mg, IQR = 14.05; Wilcoxon test: $Z = 5.06$, $P < 0.0001$; $N = 191$) (Fig. 1).

### Developmental durations and adult size in the remaining treatments

Complementary post-hoc analyses of the remaining treatment-pairs confirmed the overall limiting effects of low experimental nutrition on development and growth (Table 3). Females in the Low-Low and High-Low treatments took much longer to mature and were still very much smaller than females in the two other treatments (Fig. 2, Table 3).

Males responded differently from females, as they took longer to reach adulthood and matured at smaller size in the Low-Low treatment compared to High-Low males (Fig. 2, Table 3). Further different from females, males showed intermediate developmental durations and adult size in the High-Low and Low-High treatments relative to Low-Low and High-High males (Fig. 2), indicating less adverse effects of late low feeding conditions in males. In both sexes, High-High conditions resulted in the shortest developmental duration as well as the largest adult size (Fig. 2).

### Effects of Treatment, Sex, and Family lineage on developmental duration and adult size

We used linear mixed models to test whether phenotypic variation regarding developmental duration and adult body size can be attributed, in part, to family-specific plastic responses and to ascertain overall sex differences with respect to development and growth. The models were specified with Treatment, Sex, and Family lineage as well as the Treatment-by-Sex and Treatment-by-Family lineage interactions as explanatory variables (Start date was included as a random effect to account for slight photoperiod changes; see methods). Highly significant effects of both interactions confirmed sex-specific development and family-specific plasticity (Table 4).

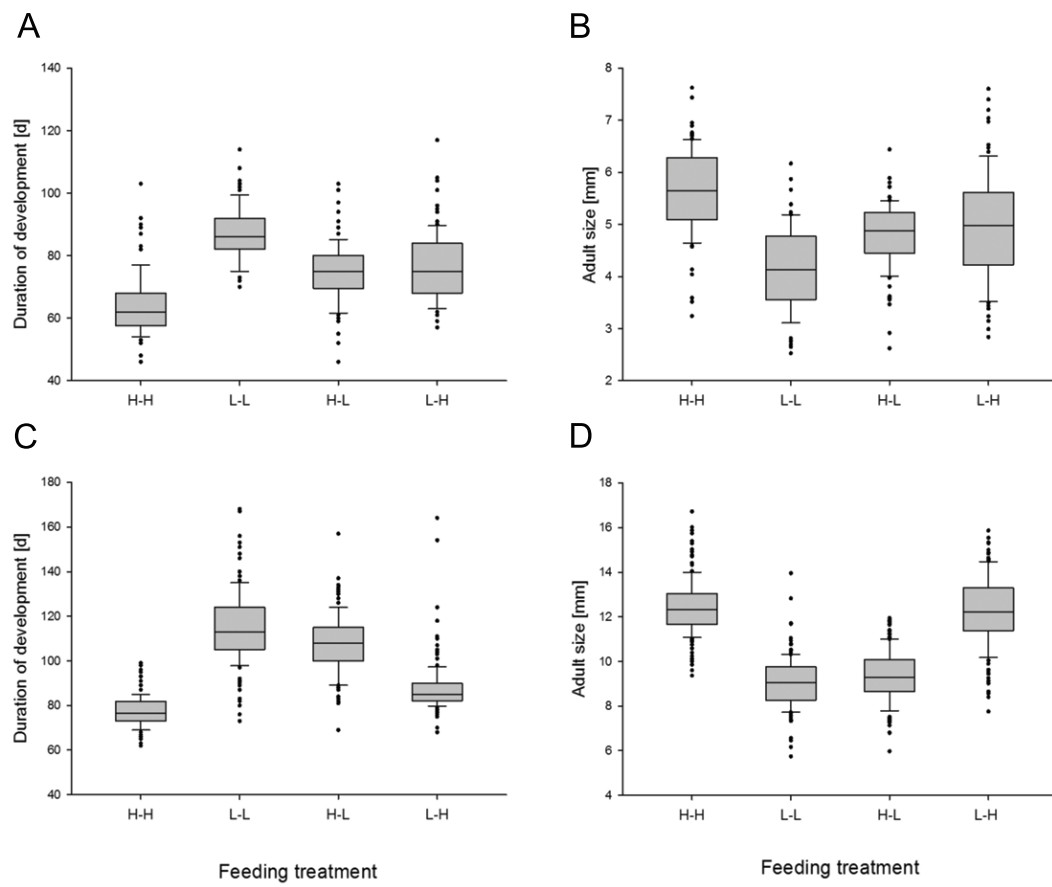

**Figure 2** **Effects of feeding treatments on the duration of development and adult size in *Nephila senegalensis*.** Top row (A, B): Males. Bottom row (C, D): Females. Treatments were High-High (H-H), Low-Low (L-L), High-Low (H-L), and Low-High (L-H).

**Table 4** **Effects of interactions between Treatment, Family lineage, and Sex on total duration of development and adult size.** Results derive from linear mixed models including the variable Start date (the time of allocation of hatchlings to individual treatments) as a random effect to account for a potential influence of photoperiod on development. Developmental durations were log-transformed. Significant *P*-values are shown in bold.

| Explanatory variable | Total duration of development | | Adult body size | |
|---|---|---|---|---|
| | *F* | *P* | *F* | *P* |
| Treatment | 339.96 | **<0.0001** | 343.43 | **0.007** |
| Family lineage | 10.3 | **<0.0001** | 2.31 | **0.033** |
| Sex | 821.48 | **<0.0001** | 7219.27 | **<0.0001** |
| Treatment * Family lineage | 3.14 | **<0.0001** | 1.94 | **0.007** |
| Treatment * Sex | 27.54 | **<0.0001** | 115.98 | **<0.0001** |

## Post-maturation implications
### Life-time fecundity and hatching success

Females originating from the Low-High treatment delayed maturation, but achieved a similar adult body size and weight as High-High females through catch-up growth (see above). We predicted equivalent fecundity in Low-High females compared to High-High females and tested differences between feeding treatments High-High, Low-High, and Low-Low (we did not include High-Low females; High-Low and Low-Low females did not differ significantly in developmental duration and size; see Table 2). Fecundity was measured as the total number of eggs (comprising hatched and undeveloped eggs) produced by each female during her entire reproductive lifespan (Life-time fecundity, LTF). The mean number of clutches was $2.82 \pm 0.25$ (range 1–6) and did not differ significantly between treatments (pairwise Tukey-Kramer HSD tests: $P > 0.1$; $N = 28$). Females originating from the Low-High treatment achieved the highest LTF of all treatments (LTF L-H = $2832.8 \pm 448.02$; $N = 10$), producing more eggs than the High-High females (LTF H-H = $2071.64. \pm 320.12$, $N = 11$) and about twice as many as females originating from the Low-Low treatment (LTF L-L = $1343.86 \pm 262.65$; $N = 7$). A linear model showed that the interaction between Treatment and the number of clutches explained a large proportion of variation in LTF ($F = 4.154$, $P = 0.03$, adjusted $R^2 = 0.83$). The model suggested that LTF increased over a series of clutches similarly in Low-High and High-High females, whereas Low-Low females were unable to achieve an equivalent increase of fecundity (Fig. 3). A Tukey-Kramer HSD test performed on model least square means showed significant differences in LTF between the Low-Low treatment and both other treatments (pairwise comparisons: L-H–H-H: $P = 0.264$, $N = 21$; L-H–L-L: $P = 0.0002$, $N = 17$; H-H–L-L: $P = 0.005$, $N = 18$).

Treatment effects on LTF did not correspond to absolute hatching success in our study. Although the total number of hatchlings produced differed considerably between treatments, variation was high and the differences were not significant (Number of hatchlings H-H: mean = $962.45 \pm 290.47$, median = 701, IQR = 2,287; L-L: mean = $421.57 \pm 133.05$, median = 528, IQR = 738; L-H: mean = $1,726.2 \pm 364.68$, median = 2,138, IQR = 2,070.75; Kruskal-Wallis test: $\chi^2 = 4.95$, $P = 0.084$; $N = 28$). We also asked if feeding regimes affected relative hatching success (i.e., the proportion of normally developed hatchlings and undeveloped eggs), but again there were no significant differences between treatments (Proportion hatched H-H: mean = $46.9 \pm 9.86$, median = 57.58, IQR = 71.81; L-L: mean = $38.01 \pm 12.76$, median = 35.31, IQR = 53.1; L-H: mean = $55.41 \pm 10.41$, median = 56.56, IQR = 49.19; Kruskal-Wallis test: $\chi^2 = 1.5$, $P = 0.47$; $N = 28$).

A post-hoc test revealed a positive correlation between copulation duration and the total number of hatchlings ($F = 18.97$, $P < 0.001$, $N = 28$, $R^2 = 0.42$). In nature, female *N. senegalensis* are polyandrous and females have been shown to mate repeatedly in other studies (*Neumann & Schneider, 2015*; *Schneider & Michalik, 2011*). In this study, single copulations were probably insufficient to fertilize all eggs produced by a female.

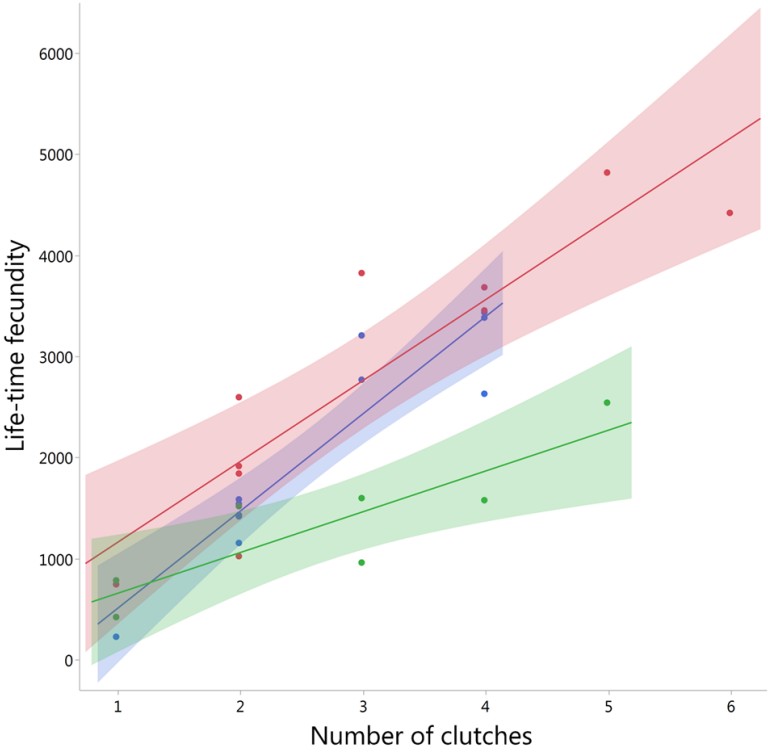

**Figure 3** **Effects of feeding treatments on life-time fecundity (LTF) in *Nephila senegalensis*.** The interaction between Treatment and the number of clutches produced explained life-time fecundity in a linear model (adjusted $R^2 = 0.83$). Treatments were High-High (blue), Low-High (red), and Low-Low (green). Shaded areas indicate 95% confidence intervals.

### *Post-maturation longevity*

We tested whether a period of juvenile food restriction and subsequent catch-up growth reduced the study animals' post-maturation lifespan, but found no significant effects in both sexes (Male adult lifespan H-H: mean $= 148.08 \pm 7.85$ days, median $= 144$ days, IQR $= 46$; L-L: mean $= 169.91 \pm 9.98$ days, median $= 178$ days, IQR $= 75$; L-H: mean $= 150.98 \pm 5.77$ days, median $= 147$ days, IQR $= 34$; H-L: mean $= 147.31 \pm 8.53$ days, median $= 150.5$ days, IQR $= 38$; Kruskal-Wallis test: $\chi^2 = 4.83$, $P = 0.185$; $N = 137$; Female adult lifespan H-H: mean $= 61.55 \pm 4.09$ days, median $= 60$ days, IQR $= 55$; L-L: mean $= 72.1 \pm 4.61$ days, median $= 69.5$ days, IQR $= 40$; L-H: mean $= 57.75 \pm 3.9$ days, median $= 54$ days, IQR $= 34.5$; H-L: mean $= 63.08 \pm 5.46$ days, median $= 61$ days, IQR $= 38$; Kruskal-Wallis test: $\chi^2 = 4.67$, $P = 0.198$; $N = 251$).

## DISCUSSION

Female *Nephila senegalensis* reared under food restriction in early development (Low-High females) used catch-up growth to counteract the restriction period and attain an adult body size and mass similar to those reared under constant food abundance (High-High females). As expected, Low-High females delayed maturation to fully compensate previous growth

deficits. Size compensation had to be charged against an average developmental delay of 9–12 days; extending development by approximately 23% compared to High-High females. This additional investment in time and growth enabled Low-High females to achieve a life-time fecundity (LTF) equivalent to High-High females; substantially exceeding LTF in constantly food-restricted females (Low-Low females). These findings further corroborate strong fecundity selection on large female size in *Nephila*. Consistent with our predictions, males did not implement catch-up growth as efficiently, showing incomplete compensation of body size. Although Low-High males delayed their development by 10–14 days, extending development by approximately 37% compared to High-High males, they matured significantly smaller than the latter. The divergence between the sexes likely reflects generally weaker selection on large male size and hence compensatory ability, but may also indicate a stronger trade-off between timely maturation and growth in males. Juvenile food restriction and compensatory development did not affect the post-maturation lifespan in either sex.

The benefits of catch-up growth have to be offset against costs of a delayed development, which certainly affect fitness under natural conditions. Environmental parameters, such as temperature, food abundance and weather conditions, may generally fall off in quality with the season approaching its end (*Hut et al., 2013*). This constitutes an 'end-of-season penalty' for late maturing individuals in semelparous species (*De Block, McPeek & Stoks, 2008*; *Higgins, 2000*). Determinate growth and annual life cycles eliminate the opportunity to optimize body size after sexual maturation and to increase fitness in future reproduction, forcing animals into a narrow time frame in which to grow and to reproduce.

In *N. senegalensis*, females produce long-lasting egg sacs which overwinter and hatch in the following spring. The spiders generally mature in late summer and early autumn to experience a relatively short reproductive period that declines with increasingly adverse weather conditions in late autumn (R Neumann, pers. obs., 2011, 2012; J Schneider, pers. obs., 2003, 2008, 2017). Such strong seasonality generates several trade-offs to cope with. For instance, a prolonged pre-maturation development entails a higher predation-risk (*Blanckenhorn, 2000*), but at the same time, large *Nephila* females outgrow the risk of being preyed upon by various invertebrate predators and parasitoids (*Chase, 1999*; *Higgins, 2002*). However, they may become more attractive to larger vertebrate predators. To make use of large body size in terms of fecundity also requires an increased amount of food and more time to produce the large numbers of eggs that can potentially be laid in multiple clutches (*Higgins, 2000*; *Neumann & Schneider, 2015*). Therefore, the adaptive significance of catch-up growth may vary between environments, for example, depending on predation pressure and the level of physical disturbance caused by extreme weather events (*Higgins, 2000*). Females are hence expected to integrate environmentally-cued information into implementing catch-up growth.

The prospects of fitness optimization through catch-up growth seem to be more limited in males, which showed less growth compensation despite a significant developmental delay. In *Nephila*, body size has often been shown to play a role in male-male competition (*Kuntner & Elgar, 2014*), but the relationship between physical dominance and increased reproductive success has been oversimplified in the past. Indeed, large males may

successfully execute their physical strength in specific competitive settings, e.g., in mating contests involving multiple males (*Rittschof, 2010*). On the other hand, there is evidence from both experimental work (*Neumann & Schneider, 2015*; *Schneider & Elgar, 2005*) and theoretical modelling (*Rittschof et al., 2012*) that small and medium-sized males adopt alternative mating strategies that balance overall paternity in competition with large rivals. Furthermore, the modelling approach suggested a decrease of average male reproductive success over the course of the season (*Rittschof et al., 2012*). This may be due to the fact that unmated females become increasingly rare and mated males may guard their females against successive competitors (*Cohn, Balding & Christenson, 1988*; *Schneider et al., 2008*). In addition, late maturing females are generally smaller and hence less fecund than early females (*Higgins, 2000*; *Miyashita, 1986*). These factors may favour protandry and amplify the trade-off between developmental time and adult size in males. The importance of a timely maturation was further supported in an experimental study, showing that male *N. senegalensis* are able to adjust the duration of their subadult instar (i.e., the last developmental stage preceding maturity) to the presence of receptive females by shifting maturation in the order of several days (*Neumann & Schneider, 2016*). Immature males use female silk (or probably silk-borne pheromones) as a cue to perceive females. Such plastic fine-tuning of life-history may increase males' chances to locate receptive females in time and avoid male-male competition, thus further relaxing selection on large male size. However, since males at least showed incomplete catch-up growth, such compensation to the minor extant should nevertheless be adaptive. Males may use moderate catch-up growth to disperse unavoidable fitness costs between both traits affected by early food restriction (the duration of development and adult body size) rather than to mend only one of them.

Sex-specific differences with respect to adaptive developmental modifications addressed in this study probably result from an evolutionary history of divergent size selection, giving way to sexual size dimorphism (SSD). Extreme reversed SSD in *Nephila* is likely facilitated by the genetic uncoupling of body size between the sexes (*Kuntner & Elgar, 2014*). The task of explaining the evolution and maintenance of extreme SSD requires identification of sex-related selection pressures. In our experiment, individuals of both sexes developed more slowly and matured smaller in the Low-Low treatment than in the High-High treatment, but apart from that, we observed considerable differences between the sexes. Males in the High-Low treatment showed similar life-history responses as Low-High males; both treatments resulting in intermediate average developmental duration and adult size relative to High-High and Low-Low males.

In females, however, the respective treatments had markedly different effects. In High-Low females, the late decrease of food supply resulted in severe limitations, as these females neither matured significantly faster nor achieved a larger size than Low-Low females. Thus, in contrast to males, females in the High-Low treatment significantly fell behind Low-High females in terms of developmental compensation, probably bearing high fitness costs. The fact that development and adult size did not differ between High-Low and Low-High males indicates less adverse effects of late-development food stress in males. A previous study addressing sex differences in *Nephila* with respect to food quantity showed

that females demand an increased food supply and especially require more food than males to reach sexual maturation (*Higgins & Goodnight, 2010*). The reduction of energetic requirements associated with small male size may therefore help to avoid fitness costs under food stress (*Blanckenhorn, Preziosi & Fairbairn, 1995*), potentially representing an important evolutionary driver to promote the uncoupling of body size between the sexes in spiders.

Although females in the Low-Low and High-Low treatments faced significant limitations regarding pace of development and adult body size, it is important to note that a large proportion of those females were still able to reach sexual maturity. Moreover, Low-Low females included in our mating trials and analyses of fecundity proved to be able to reproduce; albeit at a lowered level. Such small females that are also observed in natural populations may be able to escape reproductive failure by making 'the best of a bad job' (*Higgins et al., 2011*). These findings indicate that the general trade-off between the time invested in growth and the resulting adult size can be enforced substantially by periods of food limitation. Favourable conditions, on the other hand, may alleviate this trade-off, as males and females in the High-High treatment took the shortest average developmental duration to reach the largest average body size.

Irrespective of sex and treatment, variation of developmental duration and adult size was high. Our analyses revealed that these life-history traits were modulated by an interaction between experimental treatments (i.e., the spiders' environment) and family lineage, representing inherited genetic variation and/or maternal effects. Hence, part of the variation in these traits was caused by family-specific responses to feeding conditions, which has also been observed in other web-building spiders (e.g., *Kleinteich & Schneider, 2010*).

The ability to survive under different feeding conditions is a basic requirement to implement adaptive developmental plasticity against impending costs of food restriction. Our results suggest that juvenile *N. senegalensis* are well able to survive a period of poor feeding conditions in early developmental stages. Although early high- or low-food conditions significantly influenced the study animals' growth, these differences did not affect the chance of survival. Very low metabolic rates in general enable spiders to subsist on low quantities of food (*Foelix, 2011*; *Mayntz, Toft & Vollrath, 2003*). It is possible that phenotypic plasticity is also used to adjust metabolic rates to present conditions in order to survive food stress (*Collatz & Mommsen, 1975*). Larger juvenile stages, however, develop higher nutritional requirements to maintain all vital physiological functions, making fluctuations in food supply more dangerous (*Higgins & Goodnight, 2010*). Accordingly, we recorded a significant increase of mortality rates in spiders experiencing low-food conditions during late development.

Adaptive catch-up growth clearly bears costs of a delayed sexual maturity, but may also involve intrinsic long-term costs arising from developmental compensation, e.g., through partial elevated growth (*Hector & Nakagawa, 2012*; *Metcalfe & Monaghan, 2001*). Physiological stress can even reduce an organism's longevity (*English & Uller, 2016*; *Hornick et al., 2000*), but our feeding treatments had no effect on the spiders' adult lifespan. In contrast, permanent juvenile food restriction reduced adult longevity in another araneid, the Bridge spider, *Larinioides sclopetarius* (*Kleinteich, Wilder & Schneider, 2015*). As in

*Nephila*, female Bridge spiders delayed development and grew as large as control females; hence there was no apparent elevated growth and it remains unclear whether adverse effects on longevity resulted from dietary restrictions or the compensatory mechanism itself. Although female Bridge spiders reared under food restriction fully compensated adult size, their fecundity lagged behind control females, because they produced smaller clutches (*Kleinteich, Wilder & Schneider, 2015*). Such findings point to limitations of fitness approximations based on size measurements. It is thus important that our study could not only confirm the proposed size-increase through delayed maturation (*Higgins, 1992*), but also evaluated LTF as a direct consequence of experimentally induced developmental responses.

While our results are in accordance with our predictions, findings in other studies addressing compensatory development in size-dimorphic species diverged from predictions in whole or in part. For example, similar to our model system, fecundity-selected females in the mosquitofish, *Gambusia holbrooki*, were expected to show pronounced catch-up growth after juvenile food restriction, whereas minor catch-up growth was expected in the much smaller males whose fitness was proposed to depend less on large size. Different from predictions, however, both sexes delayed maturation and grew as large as control fish (*Livingston, Kahn & Jennions, 2014*). Male mosquitofish exhibit large size variation in nature and the authors suggest that size-related fitness consequences may depend on variable external conditions, including the social environment. Male developmental strategies may thus be influenced by population density and the intensity of male-male competition (or cues of such conditions) (*Livingston, Kahn & Jennions, 2014*). It is important, in general, to note that other environmental variables may often interact with food supply to induce specific responses, which is challenging to incorporate into experimental work (*Davidowitz, D'Amico & Nijhout, 2004*; *Stillwell & Davidowitz, 2010*).

In the pholcid spider *Pholcus phalangioides*, presenting a rare case of male-biased sexual size-dimorphism in spiders (*Uhl, 1994*), males benefit from both timely maturation and large body size by avoiding male-male competition, or by succeeding in it (*Schaefer & Uhl, 2003*). Food-restricted males were predicted to use developmental plasticity to increase body size either by delaying development or through accelerated growth. However, although males took longer to mature than control siblings, they could not catch up in terms of adult size (*Uhl et al., 2004*). Males were apparently unable to resolve the trade-off between benefits of protandry and advantages of large size under dietary restrictions (*Uhl et al., 2004*). Similarly to our findings, the observed developmental response may reflect a way of dispersing disadvantages with respect to developmental duration and adult size across both traits. It would be worthwhile, in general, to evaluate whether this kind of 'bet-hedging' through intermediate life-history traits in moderate catch-up growth is truly adaptive and how external conditions may influence developmental responses. Future studies addressing these aspects should integrate field-based knowledge regarding existing phenotypes, the social environment, and time regime, and preferably include potential interactions between multiple environmental factors that might be involved in the expression of plastic traits.

## CONCLUSIONS

Male and female *Nephila senegalensis* performed significantly different with respect to catch-up growth. Our study indicates strong fecundity selection on females, resulting in efficient growth compensation and hence prevention of fecundity-related fitness costs. Matching our predictions, catch-up growth in males did not evolve to the same capacity as in females. Relaxed selection on large male size and a stronger trade-off between costs of a delayed maturation and size-related benefits were reflected in incomplete growth compensation. Nonetheless, the moderate degree of catch-up growth in males is likely adaptive in dispersing unavoidable costs of food restriction across affected traits. The adaptive value of moderate compensatory development and the potential adjustment of such mechanisms to environmentally or socially cued conditions should be addressed in future studies.

## ACKNOWLEDGEMENTS

We are grateful to Tomma Dirks, Angelika Taebel-Hellwig, Stefanie Zimmer, Janine Helms, and Jessica Suhr for invaluable help in the rearing of study animals and data recording, Claudia Drees and Luca Neumann for graphing advice, and Matthias Foellmer and an anonymous reviewer for valuable comments on the manuscript. JMS and RN dedicate this work to the memory of our late colleague Nicole Ruppel (3.10.1978–24.2.2016).

### Funding

This work was supported by Hmb NFG-Scholarship (Universität Hamburg) and the German Science Foundation (DFG)(SCHN561/9-1 to Jutta M. Schneider). The funders had no role in study design, data collection and analysis, decision to publish, or preparation of the manuscript.

### Grant Disclosures

The following grant information was disclosed by the authors:
Hmb NFG-Scholarship.
German Science Foundation (DFG): SCHN561/9-1.

### Competing Interests

The authors declare there are no competing interests.

### Author Contributions

- Rainer Neumann analyzed the data, wrote the paper, prepared figures and/or tables, reviewed drafts of the paper.
- Nicole Ruppel conceived and designed the experiments, performed the experiments.
- Jutta M. Schneider conceived and designed the experiments, contributed reagents/materials/analysis tools, wrote the paper, reviewed drafts of the paper.

## Data Availability

The raw data has been supplied as a Data S1.

## Supplemental Information

Supplemental information for this article can be found online at http://dx.doi.org/10.7717/peerj.4050#supplemental-information.

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
