# Peer review of "Fitness implications of sex-specific catch-up growth in Nephila senegalensis, a spider with extreme reversed SSD"

_PeerJ, doi:10.7717/peerj.4050_

## Round 0.1 · original submission · Major Revisions

Please revise your manuscript, paying close attention to the comments of the reviewers. It is likely that your revised manuscript will be returned to at least one reviewer for additional evaluation.

Reviewer 1 ·

Basic reporting

This is a well-written paper and the results are interesting.

Experimental design

This is basically a well-designed laboratory experiment. Some methods need to be described with more detail (see comments for the author)..

Validity of the findings

I am concerned whether the results obtained under the artificial laboratory conditions can be used to infer selection of life-history traits in nature (see comments for the author).

Additional comments

This is a well-written paper on a basically well-designed laboratory experiment and the results are interesting. However, I am concerned whether the results obtained under the artificial laboratory conditions can be used to infer selection of life-history traits in nature.

The cost of delayed maturation on fitness, in which female spiders that mature slower in the field are more likely to be eaten by predators before they reproduce, cannot be assessed in a laboratory experiment. Although this cost is briefly mentioned in the Discussion, I am not convinced that it is unimportant in nature. Although large spiders may be too big to be eaten by invertebrate predators, they may be more attractive to larger vertebrate predators. Other factors like physical disturbance by storms may also add to the cost of delayed maturation. A more comprehensive view is that delayed reproduction at large size may be advantageous in a low-predation, low-disturbance environment but non-adaptive in a high-predation, high-disturbance environment.

The nutritional value of the flies fed to the spiders needs to be addressed. Drosophila may not contain enough nutrients for spiders to grow optimally and reach maturity. Although the methods are not clear, it seems that spiders in the low-food conditions were fed only or mostly Drosophila, whereas those in the high-food conditions may have received more of a mixture of Drosophila and Calliophora. Could the increased nutritional value of a mixed diet influenced the results? In particular, if males in the Low-High treatment were fed only or mostly Drosophila (because they were too small to capture Calliophora), this might have impeded their catch-up growth potential, whereas females in the Low-High treatment probably received a mixed diet during the second half of the experiment.

The first section of the methods states that the spiders were housed in small plastic cups. This may have been alright for small juveniles, but the cups would have been too small for larger individuals. Hopefully, at some point the larger spiders were transferred to larger containers, but this is not stated.

·

Basic reporting

The ms is well written and all relevant raw data are available. The discussion is too spider-centered. There is a large body of literature on sex-specific growth in insects which is not sufficiently cited (e.g. papers by Stillwell & Davidowitz).
Detailed comments:
L246: These headings are awkward. With regards to the analysis, why did you not model the interactive effects of family and treatment here? Also, why is the sex effect not explicitly tested? See comment below.
L304-305: Again, why is not sex included as a predictor, especially in the interaction?
L313-314: But High-Low and Low-Low might still differ with regards to fecundity.
L317-318: Why are size or mass not included as covariates?
L377: Please inform us of the phenology of N. senegalensis.
L417: ...PROBABLY result from an…
L418: …is VERY LIKELY facilitated…
L458: Detecting variation in growth responses due to additive genetic variation or maternal effects in a population does not equate to a "range of developmental strategies".
L514: You do not formally test for sex-specific growth responses or for any interactive effects of sex.

Experimental design

The study can address the research question. The rearing effort is laudable. It is very painstaking work to rear that many orb-weavers.
There are design limitations; i.e. growth was not measured for each instar, which was probably prohibitive logistically. Hence our understanding of sex-specific growth remains pretty basic.

Validity of the findings

In this study, the sex-specific effect of food limitation on growth and size at maturity was measured. That is, the results provide information on sex-specific growth mechanisms, with implications for possible sex-specific selection processes. Hence, any discussion of patterns of natural and sexual selection must be speculative and should be treated as such. However, the discussion reads as if the study provides direct insight into selection processes, which it cannot. Thus, I recommend toning down the discussion throughout.
The statistical approach of multiple pairwise comparisons falls short of providing a comprehensive analysis. It is quite peculiar that the study aims to identify sex-specific growth responses to food limitation during early and late stages of development, but no model is built with sex as a predictor or part of an interaction with treatment. SSD is of course very pronounced in adults in this species and we don’t need stats to tell us that, but sex-specific growth responses should be tested explicitly.

---

## Round 0.2 · accepted · Accept

Thank you for your careful revisions to address the reviewers' concerns.

·

Basic reporting

All my - and the other reviewer's - comments on the first version of the ms have been adequately and nicely addressed. I have have no further comments.

Experimental design

All my - and the other reviewer's - comments on the first version of the ms have been adequately and nicely addressed. I have have no further comments.

Validity of the findings

All my - and the other reviewer's - comments on the first version of the ms have been adequately and nicely addressed. I have have no further comments.

Additional comments

The paper is now in great shape and I recommend publication.

Best wishes,

Matthias Foellmer